# The Emerging Role of LHb CaMKII in the Comorbidity of Depressive and Alcohol Use Disorders

**DOI:** 10.3390/ijms21218123

**Published:** 2020-10-30

**Authors:** Chaya Shor, Wanhong Zuo, Jean D. Eloy, Jiang-Hong Ye

**Affiliations:** Department of Anesthesiology, Rutgers, New Jersey Medical School, 185 South Orange Avenue, Newark, NJ 07103, USA; tzivia54@gmail.com (C.S.); zuowa@njms.rutgers.edu (W.Z.); eloyje@njms.rutgers.edu (J.D.E.)

**Keywords:** lateral habenula, CaMKII, addiction, drugs of abuse, alcohol, depression, comorbidity

## Abstract

Depressive disorders and alcohol use disorders are widespread among the general population and are significant public health and economic burdens. Alcohol use disorders often co-occur with other psychiatric conditions and this dual diagnosis is called comorbidity. Depressive disorders invariably contribute to the development and worsening of alcohol use disorders, and vice versa. The mechanisms underlying these disorders and their comorbidities remain unclear. Recently, interest in the lateral habenula, a small epithalamic brain structure, has increased because it becomes hyperactive in depression and alcohol use disorders, and can inhibit dopamine and serotonin neurons in the midbrain reward center, the hypofunction of which is believed to be a critical contributor to the etiology of depressive disorders and alcohol use disorders as well as their comorbidities. Additionally, calcium/calmodulin-dependent protein kinase II (CaMKII) in the lateral habenula has emerged as a critical player in the etiology of these comorbidities. This review analyzes the interplay of CaMKII signaling in the lateral habenula associated with depressive disorders and alcohol use disorders, in addition to the often-comorbid nature of these disorders. Although most of the CaMKII signaling pathway’s core components have been discovered, much remains to be learned about the biochemical events that propagate and link between depression and alcohol abuse. As the field rapidly advances, it is expected that further understanding of the pathology involved will allow for targeted treatments.

## 1. Introduction

Alcohol use disorders (AUDs) is the medical diagnosis given to individuals who are suffering from severe problem drinking. AUDs are chronic relapsing brain disorders characterized by an impaired ability to stop or control alcohol use despite adverse social, occupational, or health consequences. AUDs are a severe problem in the United States. According to the National Institute of Health Alcohol Facts and Statistics, in 2018, in the United States, AUDs affect an estimated 15 million people, 5.8 percent or 14.4 million adults (ages 18 and older). This includes 9.2 million men and 5.3 million women. AUDs also affect an estimated 401,000 adolescents between the ages of 12 and 17 (National Institute on Alcohol Abuse and Alcoholism (NIAAA): Understanding the impact of alcohol on human health and well-being). 

In 2018, 26.45 percent and 6.6 percent of adults engaged in binge drinking or heavy drinking, respectively. An estimated 88,000 people (62,000 men and 26,000 women) die from alcohol-related causes each year, making alcohol the third leading preventable cause of death in the United States (Centers for Disease Control and Prevention. Alcohol and Public Health: Alcohol-Related Disease Impact). In 2014, 9967 deaths (31 percent of overall driving fatalities) were due to alcohol-impaired driving (Traffic Safety Facts CrashStats. Report No. DOT HS 812 219, Washington, DC: National Highway Traffic Safety Administration, 2015). AUDs are a severe economic burden on society. In 2010, alcohol misuse cost the United States $249 billion [1]. Three-quarters of the total cost of alcohol misuse is related to binge drinking [1]. AUDs seriously affect family life. According to a 2012 study, more than 10 percent of children in the U.S. live with a parent with alcohol struggles.

AUDs are also a severe problem worldwide. According to the World Health Organization (WHO), in 2012, the harmful use of alcohol caused 3.3 million deaths globally, or 5.9 percent of all global deaths (7.6 percent for men and 4.0 percent for women) (WHO Global status report on alcohol and health, 2018). In 2014, the WHO reported that alcohol contributed to more than 200 diseases and injury-related health conditions, most notably, Diagnostic and Statistical Manual of Mental Disorders-4 (DSM-IV) alcohol dependence, liver cirrhosis, cancers, and injuries. In 2012, 5.1 percent of the burden of disease and injury worldwide (139 million disability-adjusted life years) was attributable to alcohol consumption (WHO Global status report on alcohol and health, 2018). Globally, alcohol misuse was the fifth leading risk factor for premature death and disability in 2010. It was the primary risk factor among people between the ages of 15 and 49 (WHO Global status report on alcohol and health, 2018). In the age group of 20–39 years, 25 percent of the total deaths were attributable to alcohol consumption (WHO Global status report on alcohol and health, 2018).

In the same vein, depressive disorders (DDs), including major depressive disorders, are among the most prevalent neuropsychiatric disorders that can interfere with patients’ functioning [2,3]. DDs are mood disorders characterized by sadness severe enough or persistent enough to interfere with function and often decreased interest or pleasure in activities. Individuals who suffer from DDs may have trouble doing normal day-to-day activities, think their lives are meaningless, and even engage in suicide ideations or behaviors. The exact cause of DDs is unknown, but it may involve a combination of heredity, changes in neurotransmitter levels, altered neuroendocrine function, and psychosocial factors.

AUDs often occur with other psychiatric conditions and this dual diagnosis is called comorbidity. This pattern of comorbidity adversely affects the prognosis, course, and treatment of both DDs and AUDs. High severity in one of these disorders is associated with high severity in another condition. Alcohol dependence prolongs the course of depression and increases the risk of suicidal symptoms and behaviors. DDs invariably contribute to the development and worsening of AUDs [4]. Patients with depression and AUDs are at increased risk of relapse to heavy drinking. However, the mechanisms underlying this association are not fully understood. This gap in our knowledge prevents us to find out a better treatment strategy for these comorbidities. Professionals working with patients suffering from these comorbidities face unique and challenging dilemmas about providing the best treatment to address both conditions. Despite the growing interest in this issue, relatively few clinical studies have tested treatments for this patient population [5]. This highlights the need to understand the etiology of the disorders and develop an effective treatment regimen. 

Although many factors can contribute to the comorbidity of AUDs and DDs [6], the molecular interplay will be discussed in depth. Recently, the lateral habenula (LHb) has emerged as a crucial brain region in the pathophysiology of DDs and AUDs. Fortunately, a detailed outline of the calcium/calmodulin-dependent protein kinase II (CaMKII) signaling pathway has emerged. The target of its signaling cascade in the LHb has proven beneficial for treating AUDs, especially for those with comorbid DDs [7].

## 2. The Role of the Lateral Habenula in Depressive Disorders and Alcohol Use Disorders

Before diving into CaMKII’s prominent role in AUDs and DDs, it is essential to review the role of the habenula, a small structure in the brain located posterior to the thalamus and adjacent to the third ventricle [8,9,10]. The habenula is divided into two distinct regions: the medial habenula and the LHb. The LHb has been neglected in the scientific literature because of its small size [11]. However, interest and research in the LHb have flourished in recent years due to the revelation that it becomes pathologically hyperactive in major depressive disorders [12,13,14,15,16,17,18,19]. The LHb receives relay information from the limbic forebrain and basal ganglia structures to basically all midbrain monoaminergic centers, including the serotonergic raphe and dopaminergic midbrain nuclei [8,20,21,22], controlling mood and emotions and playing a critical part in the brain’s response to reward (Figure 1). Thus, the LHb is thought to have a role in the progression of depression and drug abuse [7,13,23,24].

The ability to seek and predict reward and pleasure and avoid confrontations is crucial for an animal’s survival and well-being. The measure of the discrepancy between reward prediction and reward outcome is referred to as reward prediction error (RPE). It serves as a powerful incentive that guides approach or avoidance (go or no-go) behavior [23,25]. Positive RPE is when the actual reward exceeds the expected one and facilitates behaviors associated with reward (approach or go). Conversely, suppressed actions related to reward (avoidance or no-go) occur when the actual reward is smaller than expected, known as negative RPE [26]. 

Information about the reward state is transmitted via the outer region of the fasciculus retroflexus axon bundle that originates in the LHb to midbrain structures, particularly the rostromedial tegmental nucleus (RMTg). The RMTg is a small nucleus that consists of gamma-aminobutyric acid (GABA) cells [27], also known as the GABAergic tail of the ventral tegmental area [28]. The fasciculus retroflexus governs the release of glutamate onto GABAergic cells in the RMTg. The resulting GABA release from RMTg neurons inhibits dopamine cells in the ventral tegmental area (VTA)/substantia nigra compacta. This process allows the LHb to control dopamine levels in their target areas, such as the medial frontal cortex and nucleus accumbens (Figure 1). These structures have essential roles in a broad range of motivated behaviors and neuropsychiatric disorders [10,20,29,30,31,32,33,34,35,36,37,38]. Thus, positive RPE has been associated with the inhibition of LHb neurons and, therefore, increases in the firing of dopamine neurons. Since dopamine produces feelings of reward, drugs that induce increased dopamine levels have abuse potential [39,40,41,42,43]. In addition, correlating with a reward prediction error (the difference between expected and actual rewards), VTA dopamine cells can also signal aversion, saliency, uncertainty, and novelty [44]. In contrast, negative RPE is associated with LHb neurons’ activation and decreases in dopamine neuron firing [45]. 

Additionally, the LHb projects to the midbrain serotoninergic system. This system plays a significant role in the motivation process [46], and its dysregulation contributes to drug addiction [47] and the etiology of mood disorders [48]. Like the projections to the midbrain dopamine neurons, LHb neurons project to the raphe serotonergic neurons either directly or indirectly through the RMTg [20,27,49,50]. LHb neurons projecting to the raphe are mostly in the medial part of the LHb, and the medial raphe (MRN) receives more LHb projections than the dorsal raphe (DRN), as shown by tracing studies [49,50]. Studies using the rabies-based viral strategy and Cre driver mouse lines show that LHb neurons project to DRN/MRN serotonergic neurons and DRN GABA neurons [51,52]. Remarkably, the raphe also sends axons to the LHb. Studies using phalloidin anterograde labeling tracing techniques show that serotonergic fibers from the MRN and DRN are throughout the whole LHb [53,54]. Studies using transgenic mice expressing Cre recombinase in serotonin-positive raphe neurons show that DRN serotonin cells project to the lateral portion of the LHb [55,56], consistent with the evidence that serotonin and its transporter are in the presynaptic side in the LHb [57,58,59,60]. For an overview on how the LHb integrates serotonergic signals, please read the recent review by Tchenio and colleagues [61].

Neurons in the LHb are almost uniformly glutamatergic [62,63]. The LHb inhibits the brain’s reward centers, including the dopaminergic VTA [21,38,64] and the serotonergic DRN [20], either via the direct projection to local interneurons within the VTA and DRN [7,27,36,38,65,66,67] or indirectly via the GABAergic relay in the RMTg [27,28,58,68,69,70] (Figure 1). The tight connection between the LHb and monoaminergic neurotransmitter systems suggests a crucial role of the LHb in neuropsychiatric diseases including DDs [18,19,34,37] and AUDs [71,72]. Accumulating evidence in rodents and humans shows that the LHb exhibits hyperactivity during depressive states [7,13,19,73,74,75,76] and alcohol withdrawal [7,71,72,77]. Conversely, a reduction of LHb hyperactivity by pharmacological or chemogenetic approaches, and deep brain stimulation or lesioning the LHb ameliorates depressive symptoms, including those in rats withdrawn from chronic alcohol [7,12,13,67,78,79,80]. Furthermore, LHb hyperactivity is involved in increased presynaptic glutamate release and upregulation of βCaMKII, which enhances the function of postsynaptic α-amino-3-hydroxy-5-methyl-4-isoxazolepropionic acid receptors (AMPARs) [13,75]. The role of CaMKII will be discussed in more detail in the later parts of this review.

Since it is a critical crossroad that influences brain responses to and encoding of aversive stimuli, such as stress, pain, anxiety, or pleasurable stimuli such as reward [23], the LHb has been correlated with major depressive disorders [13,18,26,75]. When rats and mice were treated with negative stimuli, LHb neurons were activated immediately [72,81,82]. Similarly, in vivo electrophysiological recording in monkeys showed that unexpected punitive signals such as an air puff in the eye, in addition to the omission of an expected reward, strongly excited LHb neurons [38]. Conversely, inhibiting LHb neurons suppresses depressive-like behaviors, as exhibited in animal models [7,17,71]. In addition to directly facilitating neuron activity, stress can alter the plasticity of LHb neurons, possibly contributing to long-term changes in the LHb under chronic stress, which can lead to depressive-like behaviors [26,83]. Though studies have shown enhanced synaptic activity and spike output of LHb neurons in animal models of depression, the precise molecular mechanisms by which depression leads to these changes are not yet understood [13,19,75]. Glutamate is the predominant excitatory neurotransmitter in the LHb. Potentiation of glutamate transmission causes LHb hyperactivity, as occurs in AUDs and DDs [7,13,26,71,77,84] (Figure 1 and Figure 2).

Through a series of experiments and human genome-wide studies, the LHb is linked to addiction to drugs, such as cocaine, morphine, and alcohol [23]. Drugs of abuse cause addiction because they can increase the activity of VTA dopamine neurons and dopamine levels in the target areas, including the prefrontal cortex, nucleus accumbens (NAc), and striatum [40,85,86,87]. Akin to withdrawal from cocaine [88], withdrawal from chronic alcohol increases the firing of and the glutamate transmission to LHb neurons (Figure 1 and Figure 2).

It is well accepted that the dopamine system, including the VTA, is involved in alcohol-seeking behavior and relapse [87,89,90]. Like other drugs of abuse, alcohol’s reinforcing properties involve enhanced dopamine neuron activity. A large body of evidence suggests that transition from recreational to compulsive drinking, characteristic of AUDs, involves neuroadaptations in the mesolimbic reward system [90,91]. Although alcohol acutely activates mesolimbic dopamine neurotransmission, withdrawal from chronic alcohol leads to substantial decrements in VTA dopamine neuron activity [92,93,94] and extracellular dopamine levels in the NAc [95,96]. It is believed that this dopamine hypofunction leads to a negative emotional state that drives drug-seeking behavior to restore dopamine to normal, drug-naive levels [96,97,98]. However, the mechanisms causing this dopamine hypofunction remain unclear. Recent evidence from serious electrophysiological, immunohistochemical, molecular, and behavioral experiments points to a significant role played by the LHb. LHb neurons’ hyperactivity during alcohol withdrawal may inhibit VTA dopamine neurons and lead to dopamine hypofunction, causing depressive symptoms such as anhedonia and anxiety [7,58,69,71,77,97,98]. These results provide strong support for a key role of the LHb in the connection between DDs, AUDs, and their comorbidities (Figure 1 and Figure 2). However, our understanding of the mechanisms by which chronic alcohol administration and withdrawal as well as depression increase LHb spike output and glutamatergic transmission is incomplete. 

## 3. CaMKII Structure and Regulation

Calcium ion (Ca^2+^) is a ubiquitous second messenger in cellular signaling [99,100]. The chief intracellular receptor and modulator of Ca^2+^ is calmodulin (CaM), a highly conserved Ca^2+^ sensor, and its primary targets are protein kinases [101]. Saturation of CaM with Ca^2+^ induces a conformational change that allows the protein to interact with and activate a wide variety of downstream target enzymes [102]. One of the primary downstream targets of CaM is a family of enzymes known as calmodulin-dependent kinases (CaM-kinases), which are vital for proper cellular function. Members of the CaM-kinase family are classified as serine/threonine kinases. They all have either a serine or threonine on their substrate P-sites, the targeted locale for phosphorylation. CaMKII, in particular, is among the most well-studied members of the CaM-kinase family [102]. CaMKII is composed of 12 subunits that are arranged into two rings, each containing six subunits, allowing for regulating and modifying the function and localization of the kinase [103]. Four isoforms of CaMKII (α, β, γ, and δ) exist in mammals, each one expressed from a separate gene [102], and the α and β isoforms are the most prevalent in the brain. The abundance of CaMKII in the nervous system makes it an unusual enzyme since catalytic molecules are typically found in lesser amounts [104]. Overall, CaMKII plays a significant role in regulating neurotransmitter secretion, receptor function, axonal transport, and structural modifications of the cytoskeleton [105,106].

CaMKII activation is dependent on calcium influx and the binding of Ca^2+^/CaM [102]. This complex binds to the regulatory region of CaMKII, producing a conformational change that not only leads to phosphorylation of target proteins, but also autophosphorylation, preventing the enzyme from reverting to its inactive conformation and decreasing the dissociation rate of bound CaM [103]. Thus, phosphorylated CaMKII acquires autonomous and Ca^2+^-independent activity and can become independent of Ca^2+^/CaM even after intracellular Ca^2+^ levels subside [107,108,109]. Functional alterations from autophosphorylation result, allowing Ca^2+^ to be translated into proper cellular responses. Autophosphorylation ramifications underlie the enzyme’s complex autoregulatory behavior, enabling it to activate at various frequencies of calcium spikes, detect calcium spikes, and behave as a molecular switch in learning and memory, a readout of synaptic activity [99]. Due to its switch-like regulatory properties, CaMKII is highly touted as a sophisticated control over so many disparate cellular functions and has been proposed to be a primary molecular component in the etiology of AUDs and DDs [7,75]. 

## 4. The Role of CaMKII in Alcohol Use Disorders

The role of CaMKII in AUDs has not been extensively studied. However, some evidence indicates that CaMKII contributes to several AUD-related behaviors (Table 1). Alcohol self-administration increases phosphorylation of GluA1 at the CaMKIIα recognition site (pGluA1- Ser831) in the central amygdala (CeA) of selectively bred alcohol-preferring P-rats as compared to behavior-matched (non-drug) sucrose controls. Intra-CeA injection of the AMPAR-positive modulator aniracetam or the cell-permeable CaMKII peptide inhibitor myristolated autocamtide-2-related inhibitory peptide (m-AIP), respectively, facilitated or inhibited alcohol self-administration [110]. Interestingly, CaMKII plays a different role in the aberrant behaviors induced by different doses of alcohol. In αCaMKII autophosphorylation-deficient αCaMKII-T286A mice, acute and subchronic administration of a low dose of alcohol (2 g/kg, intraperitoneal injection, i.p.) failed to induce locomotion, but a high dose (3.5 g/kg, i.p.) caused sedation. In αCaMKII-T286A mice, acute or subchronic alcohol administration did not change dopamine (DA) levels, measured by in vivo microdialysis, in the NAc, but enhanced serotonin (5-HT) responses in the prefrontal cortex were observed. Thus, αCaMKII autophosphorylation and the change in the DA–5-HT balance contribute to the establishment of alcohol-drinking behavior [111]. CaMKII may also act as a primary molecular mechanism that regulates relapse in alcohol addiction and cue-induced reinstatement of alcohol-seeking behavior, a hallmark behavioral pathology of addiction. In male C57BL/6J mice, reinstatement was associated with increased pCaMKII-T286 immunofluorescence reactive in specific reward- and memory-related brain regions, including the amygdala, NAc, lateral septum, mediodorsal thalamus, and piriform cortex, as compared with extinction control [112].

Importantly, CaMKII in the LHb plays a significant role in aversive behaviors during alcohol withdrawal [7,58,110]. The protein level of phosphorylated AMPAR GluA1 subunit at a CaMKII locus (pGluA1-Ser831) was higher in the LHb of Long–Evans rats withdrawn from chronic intermittent alcohol drinking compared with alcohol-naive rats. These alcohol-withdrawn rats showed clear depressive-like behaviors, measured by forced swimming test and sucrose-preferring test [7,58,77], which were mitigated by intra-LHb injection of the AMPAR antagonist quinoxalinediones 6, 7-dinitroquinoxaline-2, 3-Dione (DNQX) or the CaMKII antagonist 1-[N,O-bis-(5-isoquinolinesulphonyl)-N-methyl-L-tyrosy]-4-phenylpiperazine (KN-62), or inhibiting LHb activity with a chemogenetic tool or deep brain stimulation [7,113]. Conversely, in alcohol-naive rats, activation of LHb AMPARs induced depressive-like behaviors [7].

CaMKII also takes part in pain hypersensitivity that often occurs in alcoholics and AUD animal models, especially during abstinence [77,124,130]. Chronic pain is a significant contributor in DD etiology [131,132,133]. In rats, the 5-HT receptors (5-HTRs) expressed on LHb neurons [58,134,135,136] and at the glutamatergic synapses on LHb neurons [137] contribute to nociception. Intra-LHb injection of meta-chlorophenylpiperazine (mCPP, a 5-HT2CR agonist) and 2,5-dimethoxy-4-iodoamphetamine (DOI, a 5-HT2A/2CR agonist) increased nociceptive sensitivity in alcohol naive rats, while antagonists SB200646 (a 5-HT2CR antagonist) and ritanserin (a 5-HT2A/2CR antagonist), 5-HT reuptake blocker citalopram, or CaMKII antagonist KN-62 mitigated the elevated nociceptive sensitivity in alcohol withdrawn rats [58]. 

An important facet to consider in understanding AUDs is glutamate, the primary excitatory neurotransmitter in the central nervous system. The N-methyl-D-aspartate receptors (NMDARs) and AMPARs are the two major subtypes of ionotropic glutamate receptors and are critical targets of alcohol. The NMDAR is a well-characterized target of ethanol that has been associated with both drinking behavior [138] and withdrawal [139]. Acute alcohol application inhibits NMDAR function in several preparations including recombinant heteromeric NMDA channels, NMDARs expressed in the oocytes [140,141], hippocampal neurons [142,143], cortical neurons [144], and neurons in the ventral bed nucleus of the stria terminalis (BNST) [145]. In contrast, chronic alcohol exposure leads to an upregulation of NMDAR function in several brain regions related to reward and memory [146,147], such as the BNST [123] and the dorsal striatum [138]. Chronic alcohol exposure, both in vivo and in vitro, increases the gene expression of NMDAR subunits NR1 and NR2B and their polypeptide levels [148]. Enhanced NMDAR activity significantly increases the amount of calcium that enters nerve cells. Ca^2+^ influx through NMDARs and subsequent activation of CaMKII are critical events in the induction of long-term potentiation, learning, and memory [149,150,151,152,153]. 

Similarly, the AMPARs also play an essential role in regulating synaptic strength [154] and AUDs [7,155]. The glutamate A1 (GluA1), one of the four subunits (GluA1–4) in the AMPARs, plays a particularly important role in AUDs [7,110,125,156]. The LHb receives robust glutamatergic inputs and expresses the GluA1 subunit predominantly [157]. It is worth mentioning the significant role of AMPAR activation in removing the voltage-dependent block by Mg^2+^ of NMDARs. During bouts of synaptic activity, AMPAR-mediated depolarization of the postsynaptic membrane facilitates the activation of NMDARs [158].

In Long–Evans rats, activation of CaMKII enhances glutamatergic transmission by phosphorylating serine 831 (Ser831) on the GluA1 subunit and by promoting AMPAR insertion into the synapse [7] (Figure 2). This enables AMPARs to play a critical role in regulating synaptic strength [159]. Site-specific phosphorylation has been studied extensively and is likely to be upregulated to increase AMPAR surface expression, thereby enhancing glutamatergic transmission. This is the key to understanding synaptic plasticity and addictive disorders, including AUDs [7,160]. The activation of AMPARs in LHb neurons promotes drug-taking episodes and vulnerability to relapse [7,159].

Chronic alcohol exposure increases surface concentrations of GluA1 in the striatum, promoting alcohol self-administration by modifying reinforcement processes [156]. Additionally, phosphorylation of GluA1 at Ser831 is significantly increased in the LHb of rats undergoing alcohol withdrawal, consistent with the findings that phosphorylation at this site increases AMPAR activity, thus reinforcing alcohol-drinking and alcohol-seeking behaviors [7]. On the other hand, local pharmacological inhibition of LHb AMPAR activity or blocking CaMKII signaling alleviated the depressive-like behaviors and alcohol-drinking behavior (Figure 2) [7]. However, insights about the input-specific expression and subcellular localization of these receptors are still lacking and require further investigation. The combinatorial use of genetic tools, optogenetic strategies, and electrophysiology will be necessary to refine our understanding of the LHb connectivity. 

As mentioned, electrophysiological and behavioral evidence from rats also shows that adaptation in the serotonin (5-HT) 2 receptor-CaMKII signaling pathway contributes to the hyper-glutamatergic state, the hyperactivity of LHb neurons, and the higher nociceptive sensitivity in rats withdrawing from chronic alcohol consumption [124]. Through a combination of behavioral and molecular approaches, a study in the selectively bred alcohol-preferring (P) and alcohol-non-preferring (NP) lines of rats found that compared with NP rats, the P-rats had a higher sensitivity to mechanical stimuli and displayed depressive-like behaviors, as well as a higher level of CaMKII in the LHb, all of which were ameliorated by alcohol drinking [161]. CaMKII in the LHb plays a significant role not only in AUDs, but also in the other commonly abused drugs, such as morphine, as shown in a recent mouse study in which chronic morphine treatment resulted in CaMKII overexpression in the LHb [162].

## 5. The Role of CaMKII in Depressive Disorders

Depressive disorders (DDs) are one of the most widespread and incapacitating mental disorders, resulting in the loss of motivation and interest, feelings of despair, and the inability to feel pleasure [75]. Since LHb was recently discovered as the key brain region in the pathophysiology of depression, a surge in interest over CaMKII has uncovered its role in regulating several signal transduction pathways associated with learning and memory. Since it can remain activated beyond the timeframe of Ca^2+^ infiltration that initially stimulated molecule, CaMKII is often dubbed a “memory molecule” [102]. Stress-induced elevations in bCaMKII signaling, in particular, are related to depression because they influence the formation and retention of aversive memories [31]. 

Through a combination of electrophysiological, behavioral, and molecular approaches, the bCaMKII was identified as the key contributor for habenular hyperactivity and, thus, depressive-like behaviors. Recent studies have shown that amplification of bCaMKII function mediates increased depolarization of LHb neurons. When tested in rats and mice, overexpression of bCaMKII, but not αCaMKII, in the LHb, using viral vectors (adeno-associated virus 2 (AAV2)) strongly enhanced the synaptic efficacy and spike output of LHb neurons and produced profound depressive symptoms, including anhedonia and behavioral despair, which were reversed by downregulation of βCaMKII levels, blocking its activity or its target molecule glutamate A1 (GluR1) [75]. In contrast to chronic stress, acute stress has been shown to increase both isoforms, αCaMKII and bCaMKII [83]. Furthermore, an increase of bCaMKII enhanced the synaptic efficacy of LHb neurons, producing profound depressive-like symptoms. This could be because of elevated bCaMKII, strengthening AMPAR-mediated synaptic transmission and hyperactivity of the LHb. On the other hand, blocking CaMKII activity, or its target molecule, GluA1, reversed depressive-like symptoms [75].

Additionally, bCaMKII may regulate other channels of LHb neurons that enhance spike output. An important question that has yet to be addressed is the feature that renders βCaMKII sensitive to depressive stimuli and antidepressants [75]. This leads to speculation that βCaMKII plays a significant role in LHb neuronal function and is a key determinant of depression [75]. That facilitation of glutamate transmission in the LHb may lead to depressive symptoms [7,18,19,71,84,163] (Figure 1 and Figure 2). Further support of this idea is the finding that ketamine, an NMDAR antagonist, suppresses the depressive-like behaviors in rodent models of depression [18]. The discovery of ketamine’s rapid antidepressant effects is perhaps the most critical advance in the psychiatry field in recent history [164]. To investigate the underlying mechanisms, in a recent study, ketamine was applied into the LHb in rats, which quickly rescued depression-like behaviors. In vivo and ex vivo, electrophysiological recording found that burst firing, which was increased in the LHb of depressive rodent models, was suppressed by ketamine [18]. 

## 6. The Comorbidity of Depressive Disorders and Alcohol Use Disorders 

Alcoholics have extremely high rates of co-occurring psychological disorders, including depressive disorders (DDs) [165,166,167,168]. Compared to non-alcoholics, alcohol-dependent Americans have 1.7–3.6 times the risk for major depression, and 7.1 times for other drug addictions [165,167]. Although the reasons for the comorbidity of AUDs and DDs remain unknown, DDs are almost invariably involved in developing AUDs, thereby weakening the effects of treatment and increasing the risk of relapse [7,169]. Nearly one-third of patients who present with the major depressive disorder also suffer from AUDs [170]. Studies in the general population have shown that people with depressive symptoms have a 2–3-fold increased risk of developing AUDs [171]. It is essential to understand the overlapping neural mechanisms behind these comorbidities.

An important facet to consider in understanding depression that is concomitant with AUDs is the AMPARs. Enhanced AMPAR signaling via CaMKII interaction with the GluA1 may be one of the pathophysiological changes that influence drug-taking and drug-seeking behaviors and produce a phenotype akin to that of an alcoholic [159]. Building on this idea, in a study in alcohol-withdrawing rats, which showed clear depressive-like symptoms, AMPARs and CaMKII in the LHb were increased. Inhibition of LHb AMPARs, CaMKII activity, and LHb neuronal activity significantly mitigated depressive-like behaviors and alcohol-drinking and alcohol-seeking behaviors. Conversely, activation of LHb AMPARs induced depressive-like symptoms [7]. These results suggest that the LHb CaMKII–AMPAR signaling pathway may be a potential target and provide a new therapeutic approach not only to relieve symptoms of depression, but also to alleviate the desire for alcohol consumption [7]. Thus, LHb hyperactivity and enhanced glutamatergic transmissions during both DD and AUD states may be the cellular and molecular mechanisms underlying LHb’s role in the connection between depression and addiction. Building on the finding of the significant antidepressant effect of the NMDAR antagonist ketamine, interest in the effect of ketamine on AUDs is growing and has shown promising results. In humans, ketamine reduced harmful drinking by pharmacologically rewriting drinking memories [172]. In adult Wistar male rats, both ketamine and 2,3-Dihydroxy-6-nitro-7-sulphamoylbenzo(f)-quinoxaline, 6-Nitro-7-sulphamoylbenzo(f)-quinoxaline-2,3-dione (NBQX), (daily intraperitoneal injections of ketamine (2.5 mg/kg) and NBQX (5 mg/kg), alone or in combination) attenuated alcohol-withdrawal induced depression, measured by forced swim test [173]. These data provide further support for the connection between AUDs and DDs. Future studies on the effect of LHb application of ketamine and other NMDAR antagonists on the comorbidities of AUDs and DDs will further reveal the role of LHb in the connection between DDs and AUDs (Figure 2).

## 7. Conclusions

Depressive disorders concomitant with alcohol use disorders reinforce each other. Together, they are often challenging to treat, as relapse in one quickly leads to relapse in the other, and the symptoms overlap. As reviewed in this article, both depressive disorders and alcohol use disorders can cause LHb hyperactivity, which inhibits the activity of dopamine neurons and serotonin neurons in the midbrain reward center. The hypofunction of these midbrain neurons is believed to be a significant cause of psychological disorders. Both alcohol use disorders and depressive disorders increase glutamate transmission to the LHb neurons and, consequently, the activation of AMPARs and NMDARs, among others. The increased calcium influx through NMDARs activates CaMKII, a key molecule in the pathophysiology of both alcohol use disorders and depressive disorders. Since CaMKII in the LHb plays a prominent role in depression and addiction, it is expected that further study of the pathophysiology involved will allow for the development of targeted therapies to address the comorbidities.

## Figures and Tables

**Figure 1 ijms-21-08123-f001:**
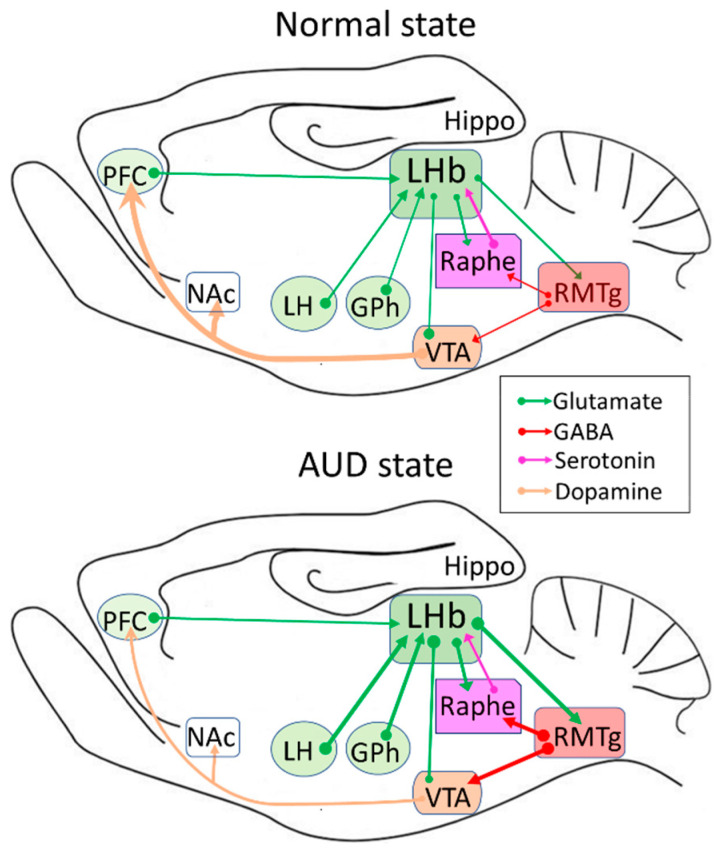
The brain network of the lateral habenula (LHb) in alcohol use disorders (AUDs) and its comorbidities. The LHb gets major inputs from the habenula-projecting globus pallidus (GPh), the prefrontal cortex (PFC), and the lateral hypothalamus (LH) and sends projections to dopamine neurons in the ventral tegmental area (VTA)/substantia nigra compacta (SNc) and serotonin neurons in the raphe directly or indirectly through the rostromedial tegmental nucleus (RMTg). The LHb also receives projections from the raphe. LHb’s spike output and glutamatergic transmission are increased in alcohol withdrawal rats and in animal models of depression, suggesting that the LHb plays a key role in AUDs and depressive disorders (DDs) and their comorbidities.

**Figure 2 ijms-21-08123-f002:**
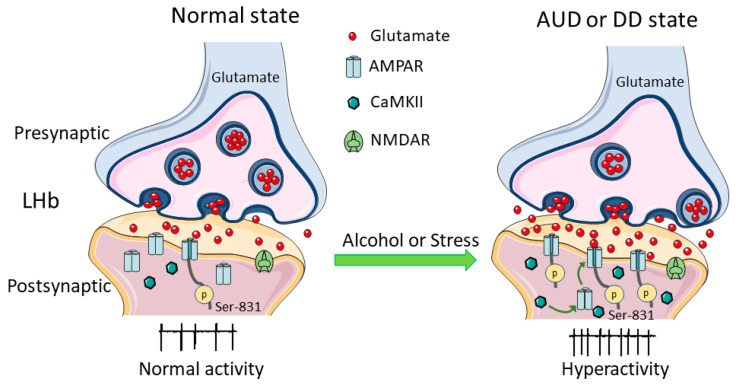
Changes of calcium/calmodulin-dependent protein kinase II (CaMKII) signaling in the LHb glutamatergic neurons in alcohol use disorders (AUDs) and depressive disorders (DDs). Withdrawal from chronic alcohol administration and stress enhance glutamatergic transmission to and the spike output of LHb neurons. This is due in part through phosphorylating serine 831 (Ser831) on the glutamate A1 (GluA1) subunit and activating the CaMKII.

**Table 1 ijms-21-08123-t001:** Evidence on CaMKII in alcohol use disorders and its comorbidities.

Brain Region	Alcohol Treatment	Tested In	CaMKII Levels	Findings	Reference
Global	2 g/kg, i.p.	pCaMKIIα-deficient, heterozygous, and WT mice	N/A	↑ EtOH’s negative reinforcing action in pCaMKIIα-deficient mice	[114]
	Three-bottle free choice			Onset of EtOH consumption delayed in pCaMKIIα-deficient mice	[111]
	2 g/kg, i.p.			pCaMKII is crucial to hippocampal DG neuron activation after EtOH exposure	[115]
PFC	Operant self-admin	C57BL/6J mice	CaMKIIα (↑)	↓ mPFC CaMKII ↑ the positive reinforcing effects of sweetened EtOH	[116]
	CIE	L-E rats (M)	CaMKIIα (↑)	pCaMKII in dorsal mPFC is associated with EtOH-induced cognitive inflexibility	[117]
	Three-bottle free choice	Mice	CaMKIIα (↓)	PFC CaMkIIα gene is linked to EtOH consumption	[118]
mPFC and NAc Sh	Self-admin (20%) (28 d)	S-D rats (M)	pCaMKII (↑) at 6 h deprivation	CaMKII mediates EtOH-WD-induced anxiety and neurochemical adaptations in mPFC and NAc	[119]
NAc Sh	CIE	Adult Wistar rats (M)	pCaMKII (↓) in high responders	↓ pCaMKII is involved in EtOH-seeking behaviors	[120]
	Self-admin (6%) (28 d)	S-D rats (M)	pCaMKII CIE (↓) 24 h WD (↑)	Activation of NMDAR1–CaMKII contributes to EtOH drinking and negative emotional states	[121]
	Drink in the dark (14 d)	Adult and adolescent C57BL/6J mice (M)	CaMKII at 24 h WD Adult (↑) Adolescent (↔)	CaMKII is positively linked to negative affective symptoms in EtOH-WD adults	[122]
BNST	0.8 g/kg, i.p. + CIE	C57BL/6J mice (M)	CaMKIIα (↓)	↓ CaMKIIα in EtOH-exposed vBNST contributes to changes in synaptic NMDAR kinetics	[123]
Amygdala, NAc, septum, thalamus, piriform cortex	Self-admin	C57BL/6J mice (M)	pCaMKII-T286 (↑)	Cue-induced reinstatement of EtOH-seeking is associated with pCaMKII-T286 in reward- and memory-related brain regions	[112]
LHb	Two-bottle free choice (8–12 w)	S-D and L-E rats (M)	CaMKII (↑) at 24 h WD	↓ CaMKII in the LHb ↓ EtOH intake and depressive symptoms	[7,124]
Amygdala	Self-admin	Adult P rats (M)	pCaMKII-T286 and CaMKIIα (↑)	↓ Acb CaMKII ↓ EtOH self-administration	[110]
	Operant self-admin (15%)	Adult C57BL/6J mice (M)	pCaMKII (↑)	EtOH drinking ↑ CaMKII in the amygdala that regulates EtOH’s positive reinforcing effects	[125]
	Home-cage drink and operant self-admin	Adolescent and adult C57BL/6J mice (M)	pCaMKIIα-T286 Adolescent (↓) Adults (↔)	Differential CaMKIIα-dependent AMPAR activation underlies age-related escalation of binge drinking	[126]
Basolateral amygdala	CIE	S-D rats (M)	pCaMKII-T286 CIE (↑), WD (↔)	CIE- and WD-induced changes in CaMKII activity contribute to ↑ GluA1R phosphorylation/trafficking	[127]
			pCaMKII-T305 CIE (↔), WD (↑)		
Hippocampal CA1 and DG	Self-admin (20%) (28 d)	S-D rats (M)	pCaMKII-T286 CIE (↓) WD (↑)	CaMKII activation in hippocampal subregions contributes to EtOH dependence	[128]
Cerebral cortex	Self-admin (10%)	Wistar rats	CaMKIIα (↑)	Pre- and post-natal EtOH exposure ↑ CaMKII levels in membrane and cytosolic fractions	[129]

↑ indicate increase, ↓ decrease, ↔ no change. AMPAR, AMPA receptor; CA1, hippocampal cornu ammonis 1; CIE, chronic intermittent ethanol exposure; DG, dentate gyrus; EtOH-WD, alcohol withdrawal; PFC, prefrontal cortex; i.p., intraperitoneal injection; L-E, Long–Evans rats; M, males; mPFC, medial prefrontal cortex; NAc Sh, nucleus accumbens shell; NMDAR, NMDA receptor; pCaMKIIα, alpha CaMKII autophosphorylation; P rats, alcohol-preferring rats; S-D, Sprague-Dawley; Self-admin, self-administration; vBNST, ventral bed nucleus of the stria terminalis; WD, withdrawal; WT, wild type; the concentrations of alcohol (EtOH), and the duration of alcohol treatment are in parenthesis.

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
