# Peer review of "The Emerging Role of LHb CaMKII in the Comorbidity of Depressive and Alcohol Use Disorders"

_ijms, 2020, doi:10.3390/ijms21218123_

Round 1

Reviewer 1 Report

The authors provide a summary of connections that have been established between depressive disorders and alcohol used disorders and the lateral habenula. They also review general characteristics of calcium signaling through calmodulin and CaMKII and describe the correlations between increased CaMKII abundance and the phosphorylation of the GluA1 subunit of AMPAR in animals with chronic alcohol exposure. This correlation is supported by inhibitor studies (Ref. 8) suggesting that CaMKII is required for the phosphorylation of GluA1 and the figure presented in this review closely resembles the model published in the referenced study. The authors should take care in concluding that a direct phosphorylation GluA1 by CaMKII occurs in vivo because it is not clear that this has been established based on the literature cited. The focus of the review is on CaMKII but others have speculated about other signaling proteins that might play a role in this section of the nervous system. As in many signaling pathways the complexity of regulation expands as more signaling components are identified and uncovering all of the important regulators is an expectation of future research. The review is well written and organized.

Concerns –

Line 225 – Activation of CamKII enhances . . .  This sentence needs a reference.

Author Response

a reference has been added, as suggested 

Reviewer 2 Report

The manuscript by Shor and colleagues is an interesting and well-written review on LHb CaMKII in alcohol use disorders and its comorbidities.
I suggest the authors
i) describe better the role of 5-HT, at the moment only rapidly mentioned, while from an anatomical and pathophysiological point of view it is highly relevant.
ii) include a new figure describing the brain network that includes LHb in alcohol use disorders and its comorbidities
iii) a table summarizing the evidence on CaMKII in alcohol use disorders and its comorbidities would help the reader

Author Response

Reviewer 2.

The manuscript by Shor and colleagues is an interesting and well-written review on LHb CaMKII in alcohol use disorders and its comorbidities.
I suggest the authors
i) describe better the role of 5-HT, at the moment only rapidly mentioned, while from an anatomical and pathophysiological point of view it is highly relevant.

As suggested, the role of 5-HT has been better described. Please see lines 122-136.

ii) include a new figure describing the brain network that includes LHb in alcohol use disorders and its comorbidities

We have added a new figure describing the brain network that includes LHb in alcohol use disorder and its comorbidities (New figure 1).

iii) a table summarizing the evidence on CaMKII in alcohol use disorders and its comorbidities would help the reader

We have added a table summarizing the evidence on CaMKII in alcohol use disorders and its comorbidities (Table 1)

Reviewer 3 Report

The authors widely explain CaMKII role in the depression and alcohol abuse. However, the explanations were not enough to get knowledge CaMKII in link between depression and alcohol abuse. The also should define the difference of alcohol toxicity and alcohol abuse behaviors. The depression is seemed to associate with the alcohol toxicity or alcohol depletion.

Major comments.

  1. Lines 191-203: The explanation of CaMKII structure is too general. The authors should focus on the characteristic for abuse-related structure properties such as autophosphorylation changed by drug abuse.
  2. Emerging evidence dictates the importance of CaMKII on AUDs-associated behaviors [8,89]. Please described in detail the results changed CaMKII activity and its regions in AUDs-associated behaviors. The authors should define the AUDs-associated behaviors to separate the toxic effects of
  3. Lines 209-217: The authors should specify the regions or neuronal circuits changed by alcohol abuse and specify the animal.
  4. Line 25: Since LHb was recently discovered as the key brain region in the pathophysiology of depression When tested in rats, overexpression of βCaMKII, but not αCaMKII in the LHb induced depressive-like behaviors [50]. Please describe the methods and results in detail.
  5. The resulting GABA release from RMTg neurons inhibits dopamine cells in the ventral tegmental area (VTA)/substantia nigra compacta. This process allows the LHb to control dopamine levels in their target areas, such as the medial frontal cortex and nucleus accumbens. These structures have essential roles in a broad range of motivated behaviors and neuropsychiatric disorders [27]. Please add more references to get this conclusion.
  6. Neurons in the LHb are almost uniformly glutamatergic [35,36]. The LHb inhibits the brain’s reward centers, including the dopaminergic VTA[37–39] and the serotonergic dorsal raphe nucleus (DRN)[40], either via the direct projection to local interneurons within the VTA and DRN [25,39,41–43] or indirectly via the GABAergic relay in the RMTg [25,26,44–47]. Please show the scheme to explain the circuit for alcohol abuse.
  7. Since CaMKII in the LHb plays a prominent role in depression and addiction, it is expected that further study of the pathophysiology involved will allow for the development of targeted therapies to address the comorbidities. The authors should add the references and explain the role of LHb to connect between depression and addiction. The readers overall could not understand the relationship between depression and addiction. The authors may focus only drug addiction not depression because depression is related to alcohol toxicity or alcohol depletion effect. If not please explain the effects of anti-depressive drugs for alcohol drug abuse and withdrawal. 

Author Response

Reviewer 3.

The authors widely explain CaMKII role in the depression and alcohol abuse. However, the explanations were not enough to get knowledge CaMKII in link between depression and alcohol abuse. The also should define the difference of alcohol toxicity and alcohol abuse behaviors. The depression is seemed to associate with the alcohol toxicity or alcohol depletion.

Major comments.

Lines 191-203: The explanation of CaMKII structure is too general. The authors should focus on the characteristic for abuse-related structure properties such as autophosphorylation changed by drug abuse.

The explanation of CaMKII structure has been revised and its now more focus on the characteristic for abuse-related structure properties such as autophosphorylation changed by drug abuse.

Emerging evidence dictates the importance of CaMKII on AUDs-associated behaviors [8,89]. Please described in detail the results changed CaMKII activity and its regions in AUDs-associated behaviors. The authors should define the AUDs-associated behaviors to separate the toxic effects of

The revision has provided the details of the results changed CaMKII activity and its regions in AUDs-associated behaviors (Table 1).

Lines 209-217: The authors should specify the regions or neuronal circuits changed by alcohol abuse and specify the animal.

The revision has provided information about the specific regions or neuronal circuits changed by alcohol abuse and the animal. (see lines 240-275, and Table 1)

Line 25: Since LHb was recently discovered as the key brain region in the pathophysiology of depression When tested in rats, overexpression of βCaMKII, but not αCaMKII in the LHb induced depressive-like behaviors [50]. Please describe the methods and results in detail.

The methods and results have been described in detail (see lines 370-373).

The resulting GABA release from RMTg neurons inhibits dopamine cells in the ventral tegmental area (VTA)/substantia nigra compacta. This process allows the LHb to control dopamine levels in their target areas, such as the medial frontal cortex and nucleus accumbens. These structures have essential roles in a broad range of motivated behaviors and neuropsychiatric disorders [27]. Please add more references to get this conclusion.

More references have been added (Lines)

Neurons in the LHb are almost uniformly glutamatergic [35,36]. The LHb inhibits the brain’s reward centers, including the dopaminergic VTA[37–39] and the serotonergic dorsal raphe nucleus (DRN)[40], either via the direct projection to local interneurons within the VTA and DRN [25,39,41–43] or indirectly via the GABAergic relay in the RMTg [25,26,44–47]. Please show the scheme to explain the circuit for alcohol abuse.

A scheme explaining the circuit for alcohol abuse has been added (Fig 1)

Since CaMKII in the LHb plays a prominent role in depression and addiction, it is expected that further study of the pathophysiology involved will allow for the development of targeted therapies to address the comorbidities. The authors should add the references and explain the role of LHb to connect between depression and addiction. The readers overall could not understand the relationship between depression and addiction. The authors may focus only drug addiction not depression because depression is related to alcohol toxicity or alcohol depletion effect. If not please explain the effects of anti-depressive drugs for alcohol drug abuse and withdrawal.

Information of the effects of anti-depressive drugs for alcohol abuse and withdrawal has been added in the revision (lines 420-425)

Round 2

Reviewer 3 Report

I have no further concerns in the revised Ms..